# U-Net Architecture for Prostate Segmentation: The Impact of Loss Function on System Performance

**DOI:** 10.3390/bioengineering10040412

**Published:** 2023-03-26

**Authors:** Maryam Montazerolghaem, Yu Sun, Giuseppe Sasso, Annette Haworth

**Affiliations:** 1School of Physics, The University of Sydney, Sydney, NSW 2006, Australia; 2Radiation Oncology Department, Auckland City Hospital, Auckland 1023, New Zealand; 3Faculty of Medical and Health Sciences, University of Auckland, Auckland 1010, New Zealand

**Keywords:** prostate cancer, prostate segmentation, U-Net, mp-MRI, loss function, medical imaging

## Abstract

Segmentation of the prostate gland from magnetic resonance images is rapidly becoming a standard of care in prostate cancer radiotherapy treatment planning. Automating this process has the potential to improve accuracy and efficiency. However, the performance and accuracy of deep learning models varies depending on the design and optimal tuning of the hyper-parameters. In this study, we examine the effect of loss functions on the performance of deep-learning-based prostate segmentation models. A U-Net model for prostate segmentation using T2-weighted images from a local dataset was trained and performance compared when using nine different loss functions, including: Binary Cross-Entropy (BCE), Intersection over Union (IoU), Dice, BCE and Dice (BCE + Dice), weighted BCE and Dice (W (BCE + Dice)), Focal, Tversky, Focal Tversky, and Surface loss functions. Model outputs were compared using several metrics on a five-fold cross-validation set. Ranking of model performance was found to be dependent on the metric used to measure performance, but in general, W (BCE + Dice) and Focal Tversky performed well for all metrics (whole gland Dice similarity coefficient (DSC): 0.71 and 0.74; 95HD: 6.66 and 7.42; Ravid 0.05 and 0.18, respectively) and Surface loss generally ranked lowest (DSC: 0.40; 95HD: 13.64; Ravid −0.09). When comparing the performance of the models for the mid-gland, apex, and base parts of the prostate gland, the models’ performance was lower for the apex and base compared to the mid-gland. In conclusion, we have demonstrated that the performance of a deep learning model for prostate segmentation can be affected by choice of loss function. For prostate segmentation, it would appear that compound loss functions generally outperform singles loss functions such as Surface loss.

## 1. Introduction

Multiparametric magnetic resonance imaging (mp-MRI) is increasingly being used in the computer-aided diagnosis, computer-assisted surgery and radiation therapy planning for prostate cancer [1,2]. Accurate prostate segmentation for radiation therapy treatment planning is necessary to ensure the prostate receives an adequate amount of radiation for tumor control whilst minimizing the amount of dose received by other organs, such as the bladder and rectum [3]. Manual segmentation has been shown to demonstrate a high degree of intra- and inter-variability, particularly at the base and apex of the prostate [4]. Additionally, manual segmentation is subjective, time-consuming and can be affected by level of experience. In comparison, automatic segmentation is fast and can decrease human bias and errors [5,6,7].

The U-Net [8] architecture has been successfully applied in prostate segmentation in several studies [9,10,11]. However, applying deep neural networks for this task can result in variable outcomes, as multiple factors can influence the model outcome. Firstly, performance of auto-segmentation models are highly dependent on training dataset features, quality and number of samples [9]. In particular, the small sample size typically used in prostate segmentation models makes automatic segmentation very challenging. Prostate shape and texture may vary widely between different patients and the heterogeneity of the prostate tissue presents additional challenges for automated segmentation. In addition, design and configuration of a deep-learning-based segmentation model requires careful consideration. There are many parameters and hyper-parameters that need to be optimized to achieve acceptable model performance. These include network architectures, training schedules, data pre-processing, data augmentation (if used), data post-processing, and several essential hyper-parameter tuning steps such as learning rate, batch size, number of epochs, or class sampling [2]. In addition, hardware availability for training and inference of these models should be considered in advance [12,13]. Model performance varies substantially with the training dataset’s properties and its size. Therefore, the applicability of trained public models for unseen datasets is limited [2], and training a model from scratch or retraining other models are popular solutions in medical image segmentation tasks.

One of the key parameters of deep-learning-based models that plays an important role in model training and success of the segmentation model is the *loss function*, also known as the cost function. The loss function is ultimately responsible for how the model’s weights are adjusted for optimization goals, such as minimizing region mismatches between predicted and ground truth segmentations. Various domain-specific loss functions have been proposed and applied for segmentation of the prostate and other organs to improve results for their datasets [9,14]. It can be challenging to know which loss function meets the requirements of the task, and whether the right function for a specific dataset has been chosen [14]. In the past ten years many loss functions have been proposed. Jadon [15], for example, reported the performance of thirteen well known loss functions designed for fast model convergence, and proposed a new loss function for skull segmentation from CT data. Ma et al. [14] provided a comprehensive review of twenty loss functions based on four CT-based publicly available data sets. For our study, we have chosen to complement these works with a focus on nine loss functions, applied to a single MRI-based data set sourced from an in-house study. This data set provides ground-truth prostate-gland segmentations based on whole-mount histology (rather than clinician generated segmentations which form the basis of many segmentation models). These nine loss functions are commonly used in medical image segmentation models and are intended to be representative of the many loss functions reported in the literature, and in particular, form a sub-set of those reported by Ma et al. and Jadon [14,15] with at least one loss function from each of the four categories defined in both studies and excluding those relevant to multi-class solutions that are not relevant here. Whilst there are many applications of segmentation models, our study was motivated by the need to develop a segmentation algorithm to analyze data collected as part of a clinical trial investigating the ability of quantitative multiparametric MRI to assess response to radiation therapy (ANZCTR UTN U1111-1221-9589). Our longitudinal data set generated a large amount of data that required an objective delineation of the prostate gland prior to the extraction of radiomic features to develop treatment response predictive models. As part of this study we identified a lack of comprehensive comparisons of prostate segmentation model performance using different loss functions. In this study, we compared deep-learning-based prostate segmentations of T2-weighted (T2w) MR images, using nine different loss functions for 2D U-Net with our locally acquired dataset.

## 2. Materials and Methods

**Dataset:** In vivo mp-MRI data were collected from 70 patients prior to radical prostatectomy as part of a Human Research Ethics Committee (HREC)-approved project called “BiRT” (HREC/15/PMCC125). These images were acquired using a 3T Siemens Trio Tim machine (Siemens Medical Solutions, Erlangen, Germany). The first 37 cases imaged using a standardized imaging protocol and free of major artifacts were available for analysis at the time this study was performed [16]. Prostate segmentations were generated from the whole-mount histology slides and subsequently co-registered with the mpMRI using a sophisticated co-registration framework [16]. For quality control, the co-registered prostate masks were checked against an independent annotation by an experienced radiation oncologist (GS) on the in vivo 3D T2w images using RayStation v.8 (RaySearchLabs Stockholm, Sweden). These contours were used as ground truth for automatic segmentation. Following segmentation of the entire prostate gland, each prostate volume was mathematically divided into sub-regions by thirds in the craniocaudal axis, with the most superior volume labelled “base”, the inferior volume “apex” and the central volume “mid-gland”. T2w images were acquired using a turbo spin echo sequence with two sets of resolutions. For the first set, the in-plane resolution was 0.6 mm × 0.6 mm, the inter-plane distance was 6 mm. The volumes of the first set contained between 80 and 96 slices each, with each slice resolution being 384 × 384 pixels. For the second set, the in-plane resolution was 0.8 mm × 0.8 mm, and the inter-plane distance was 0.8 mm. The volumes of the second set contained between 80 and 88 slices each, with each slice resolution being 256 × 256 pixels.

Pre-processing of the input data included bias field correction, resampling, and image normalization. The intensity range of each image was normalized using minimum and maximum intensity values of each single image before incorporation into the network. The datasets were resampled into 128 × 128 × 64 voxels. A flow chart indicating the image processing pipeline is shown in Figure 1. The full pelvic field of view was used without cropping.

**U-Net architecture and Loss Functions:** The effect of various loss functions on the performance of a basic 2D U-Net architecture [8] was investigated using the T2-weighted MR images. Loss functions were selected from traditional distribution-based and region-based categories, as well as more recent compound and boundary-based loss functions. Most of the loss functions used in this study were selected based on their suitability for use with strongly and mildly imbalanced data sets in segmentation tasks and those commonly used in medical image segmentation models [14]. These include Binary Cross-Entropy (BCE), Intersection over Union Loss (IoU), Dice Loss, combination of Dice and BCE loss functions (BCE + Dice), weighted BCE and Dice Loss (W (BCE + Dice)), Focal Loss, Tversky Loss, Focal Tversky Loss and Surface [9,14,17]. Table 1 summarizes the loss functions used in this paper with loss function definitions based on those of Ma et al. [14], with details included in Appendix B.

The U-Net architecture contains two main components: the encoder or contracting path, which extracts the features of the image by applying a stack of convolutional and max pooling layers (Figure 2, left), and the decoder or expanding path (Figure 2, right). The U-Net architecture is an end-to-end fully convolutional network (FCN) and contains only convolutional layers without any dense layers. This allowed the network to accept images of any size.

The encoder of the network used in the current study had five convolutional layers to extract high-level feature maps. In each convolutional layer, the input feature map was convolved with a set of trainable filters, kernels of size 3 × 3 and a 2 × 2 max pooling operation with a stride of 2. Max pooling operations or down-sampling reduced the feature map size by a factor of 2 in each dimension. Then, a batch normalization operation was applied, followed by rectified linear unit (ReLU) activation functions. ReLU performed the thresholding operation (*max* (*x*,0)), used to introduce nonlinearity to the trained network. The number of feature channels started at 16 for the first stage, and doubled after each stage of the decoder to 32, 64, 128, and finally 256.

A decoder reverses the operations of the encoder to recover the original input size and enable the network to perform a voxel-wise classification. Each stage of the decoder included two types of operations. Firstly, layers were up-sampled to increase the size of the feature map gradually until it reached the size of the original input image. Secondly, deconvolutional layers reduced the number of feature channels to half at each stage of the decoder to match the number of channels with the corresponding encoder layers. Features extracted from earlier stages were added to the encoder side (Figure 2) using short-circuit layers to help recover the spatial information from the convolutions in the encoder.

The U-Net model applied in this study had nine convolutional layers. Model parameters, except the loss function, were fixed for all models. The Adam optimizer [18] was selected as the optimization algorithm, with an initial learning rate α = 0.0001, a learning rate drop factor of 0.1, and a patience of 10 (meaning that the learning rate dropped by a factor of 0.1 when the validation loss did not improve for 10 epochs). The training was performed for 10,000 epochs with an early stopping strategy and a batch size of 2 to avoid overfitting. Model training was stopped when the validation loss did not improve for 10 epochs. Dropout was applied for each convolutional layer at a rate of 10% to avoid overfitting. Batch normalization was applied after each convolution layer to prevent gradient vanishing/exploding [19]. The results for each model reported the best epoch based on the validation set. The number of model parameters was 1,189,264, of which 1,187,792 were trainable. Sigmoid activation was used as the output layer for binary predictions. A threshold value of 0.5 for the probability was applied to obtain the segmentation mask, this value was found to be the optimal value that gave the highest Dice coefficient (DSC) and fewer false positives.

Five-fold cross-validation was used to validate the results [20]. For model selection, the best model was determined based on performance of the validation datasets [14]. Our proposed network was implemented in Keras v2.3.1 [21], using TensorFlow v2.0.0 [22] backend with Python. For each loss function, the network was trained by performing a five-fold cross-validation using all 37 cases from the BiRT dataset. All calculations were performed using the University of Sydney’s HPC service and GPU access, NVIDIA V100 SXM2.

**Evaluation Metrics:** Models were compared and evaluated using commonly used metrics for medical image segmentation [23]. These include the DSC, 95% Hausdorff Distance (95HD), relative absolute volume difference (Ravd), precision, and sensitivity. These metrics were selected to cover evaluations for region-based, contour-based and volume-based similarities between the ground truth and auto-segmentation output. A DSC score of 1 shows perfect agreement. The Hausdorff Distance measures the distance between the borders of the ground truth and the auto-segmentation output. Lower values of 95HD indicate a better performance of segmentation. Ravd is the difference between the total volume of the segmentation and the ground truth divided by the total volume of the ground truth. The Ravd value for a perfect segmentation is equal to zero.

## 3. Results

Table 2 provides a summary of the results of the different loss functions applied to the nine models used in this study. Figure 3 shows box plots for each of the nine models and evaluation metrics for the whole prostate. Appendix A contain boxplots for these models for the prostate mid-gland, apex, and base, respectively. Figure 4 shows DSC box plots for different parts of the prostate. The mid-gland (Figure 4C) shows a consistently high performance (except for Surface loss), followed by the base and the apex (Figure 4B,D, respectively). Table 2 shows that the Focal Tversky loss function had the highest average of DSC scores for the whole gland and the lowest standard deviation (0.74 ± 0.09). Models with IoU, Dice, Tversky and W (BCE +Dice) and BCE + Dice loss functions obtained similar DSC scores of 0.73, 0.73, 0.72, 0.71, and 0.71, respectively. A high DSC score was expected for these loss functions as they are variates of the Dice coefficient and aim to minimize this metric during the training process. Additionally, the Dice loss function and its variates perform better in class-imbalanced problems such as prostate segmentation. Models with surface and BCE loss functions had the lowest whole gland DSC, with values of 0.40 and 0.58, respectively. The maximum difference in DSC score across all models’ performance was approximately 34%.

In considering DSC scores shown in Figure 4, it can be seen that all models achieved the highest DSC score for the mid-gland (Figure 4C), which had a 20% (up to 93–94%) higher accuracy compared to the whole prostate (Figure 4A), most likely because the whole gland resembles the mid-gland, and it accounts for the majority of the prostate volume. Model performance was lower in the apex (Figure 4B) when considering all parts of the prostate and the prostate as a whole. Higher standard deviations of the DSC scores were observed for the apex from all models (Table 2).

Regarding 95HD, the best performance was achieved by W (BCE + Dice), with a value of 6.66 ± 2.82 for the whole prostate gland, followed by Tversky and Focal Tversky with values of 7.17 ± 4.21 and 7.42 ± 5.81, respectively (Table 2). The worst performing model was Surface, with a value of 13.64 ± 4.38, approximately double that of the best performing model (W (BCE + Dice). When considering the base, mid-gland, and apex, as expected, the mid-gland reported lower 95HD values, followed by the apex, with the best performance achieved by W (BCE + Dice) and Dice, respectively.

Ravd is an appropriate metric for applications with an interest in accurate volume estimation and similarity. An absolute value of Ravd approaching zero shows a better model performance. The lowest absolute values of Ravd for the whole prostate were obtained from W (BCE + Dice), BCE + Dice, Surface and Dice (0.05, 0.07, 0.09 and 0.09, respectively) and the largest deviation from a score of zero was Focal with a value of −0.25 ± 0.31 (Table 2). The standard deviations of Ravd for models with W (BCE + Dice) and BCE + Dice were small, with values of 0.31 and 0.37, respectively.

The highest sensitivity value was achieved for the whole prostate gland using Focal Tversky (80%), and the lowest using the Surface loss function (44%). Similar values of precision were achieved for all loss functions for the whole gland (69–73%), with the exception of Surface (51%). Focal Tversky, W (BCE + Dice) and Focal each have parameters which can control trade-off between false positives and false negatives (FP and FN). These parameters can be optimized based on segmentation task needs and data properties.

The surface loss function had the lowest DSC score and a higher 95HD. This model had the lowest performance considering the majority of metrics used in this study. Furthermore, models with a surface loss function required longer training times and higher numbers of iterations.

There was a pattern of improved DSC score in slices that covered a larger area of prostate, mainly in the mid-gland with cross sectional areas greater than 600 mm^2^ and less than 2100 mm^2^. This is represented in Figure 5, where the data shown is based on the prediction from the model using W (BCE + Dice) on the validation data. The same pattern is seen in all models. Figure 6 presents the box plots for all loss functions.

Within the Appendix A shows the DSC scores for individual patients for each model. Box plots of the DSC scores of all the models for each patient on the validation datasets in the five-fold cross validation are shown in Figure 7. DSC scores of models varied between patients, but for each patient the results were generally consistent across all three models (Tversky, Focal Tversky and W (BCE + Dice) (Appendix A).

Model performance was generally lower in the apex and base compared with the mid-gland. This was not surprising, as inter-observer variability has been reported to be higher in these regions [4]. However, this may be an effect of the small cross-sectional areas (Figure 5 and Figure 6). Additionally, the DSC score was lower for the slices that covered small areas or very large areas. We investigated the relationship between DSC score and prostate volume (Figure 4). No clear trend was identified, possibly due to the limited number of samples. However, in general, the model showed lower performance in DSC scores for smaller volumes in comparison to the average volume.

### Qualatative Comparison

A selection of cases representing high and low performance are shown in Figure 8 and Figure 9 for the models’ outputs using two different loss functions, Focal Tversky and W (BCE + Dice). Samples with DSC scores higher than 0.80 were considered high-performance cases, and lower than 0.70 were considered low-performance cases. Higher DSC scores were achieved, for example in patients (cases) #2, # 16, #21, and #33. Cases #3, #8, and #22 are examples of lower performance.

Segmentation results show higher DSC scores from the model with Focal Tversky for case # 33 compared to W (BCE + Dice), with values of 0.83 and 0.87, respectively. Both models failed to capture the shape of the prostate at the apex and base. However, the output of the model with Focal Tversky had a greater similarity to the shape of the prostate than W (BCE + Dice) (Figure 9, case #2). This indicates that the model using Focal Tversky was more effective in defining the prostate boundaries.

Both models failed to define the prostate boundary for cases #15 and #22, especially in the apex and base regions. From the rectum shape in case #22, it is possible that there is some gas in the rectum which can reduce the quality of the MRI image.

For case #16, the model using W (BCE + Dice), with a DSC score of 0.79, had a worse performance compared with Focal Tversky (DSC score 0.85). The segmentation output of the model with W (BCE + Dice) was rectangular in shape, which can be seen in the coronal and sagittal views (Figure 9). Shapes of the segmentation outputs from Focal Tversky had a closer shape to the prostate than those from models with W (BCE + Dice) loss function.

In general, the W (BCE + Dice) model under-estimated the prostate volume and the Focal Tversky over-estimated the volume. Examples are cases #22, #15, #8, #16, and #2 (Figure 9).

## 4. Discussion

Finding the most appropriate loss function for prostate segmentation is challenging. In this study we compared the performance of nine loss functions in a 37-patient data set. These nine loss functions were chosen as they are commonly used in medical image segmentation tasks [14]. The 37-patient data set included locally acquired data with a common imaging protocol (two resolutions) and a single MRI scanner [16] to avoid variations due to image acquisition. These data were co-registered with whole mount pathology to provide ground truth delineations of the prostate [16] in contrast to many publicly available datasets that rely on clinician-generated segmentations which are subject to interobserver variation [4]. A limitation of the generalizability of our study is the small sample size and homogeneity in the methods used to acquire the MRI data. We therefore recommend that future studies that intend to use data from a variety of sources and scanning protocols confirm the findings of our study using the methodology we describe, and consider the most appropriate metric for their evaluation. Publicly available data can be sourced from a variety of locations such as those described by Ma et al. [14], however, the purpose of our study was to remove uncertainties due to heterogeneity in data source and clinician contouring, and focus only on the relative performance of the loss functions selected for our study and a range of metrics for their evaluation. Our study found the proposed architecture performed with notable variations when different loss functions were applied. As the base and the apex of the prostate are particularly challenging to segment manually due to the lack of a clear boundary [1,17], we therefore also evaluated the performance at the mid-gland, apex, and base of the prostate independently.

Focal Tversky had the highest scores for the whole gland in terms of DSC score and sensitivity. However, W (BCE + Dice) outperformed all competing methods in precision, followed by 95HD, Ravd, and Tversky. With performance measured by the median and standard deviation, the best performance was achieved by applying W (BCE + Dice), Tversky, and Focal Tversky loss functions. However, the performance of models with Focal Tversky, Tversky, W (BCE + Dice), Dice, and IoU loss functions were very close for our dataset. Lower performance was observed using Surface loss, BCE loss and Focal loss functions. Focal Tversky and Tversky loss functions have been recommended by other researchers as returning optimal results when their parameters are set to the correct values [15]. However, for challenging medical segmentation tasks, we suggest using Focal Tversky and W (BCE + Dice), and by optimizing their parameters, the best solution can be achieved in accordance with the application requirements. The loss function parameters of W (BCE + Dice) allow the user to define the best trade-off between FNs and FPs. Additionally, Focal Tversky and W (BCE + Dice) have the advantage of adjustable parameters, which make it possible to tune the loss function based on the application requirements. For example, Focal Tversky and W (BCE + Dice) have parameters which can be tuned to address under- and over-segmentation issues that may arise with other loss functions. As a result, in the future, we plan to investigate the effectiveness of a combination of Tversky and BCE loss functions for prostate segmentation.

Lower performance was observed using Surface loss, BCE loss and Focal loss functions. All models achieved higher performance for mid-gland and lower performance in the apex and base regions. When considering model performance for individual data sets, we observed that all models had a similar performance for each image, but performance varied across the patient cohort. This may be related to patient-specific image quality, however, all models generalized the average shape of images and failed to perform well for outlier shapes.

Intuitively, it can be expected that model performance will be affected by the choice of the metric used to measure performance and the principal components driving the loss function. For example, DSC measures the overlap between two regions. If the Dice loss is used, the training process is exactly guided as the final metric, which theoretically should achieve a good performance. This can be seen in Table 2; the Dice loss achieved a consistently high DSC in the whole prostate gland (0.73) as well as the sub-volumes (0.65–0.93). In addition, the close variants of the Dice loss, including Tversky, Focal Tversky, and IoU loss, also obtained high performances (0.63–0.92), but slightly inferior to the Dice loss. For losses that are not region-based, compound losses such as BCE + Dice and W (BCE + Dice) showed relatively higher DSC (0.62–0.93) as they consist of a Dice loss component. In contrast, Surface loss (boundary-based) and BCE (distribution-based) demonstrated the lowest DSC (0.38–0.75). However, this pattern is not shown between all metrics and categories. For example, HD95 is a boundary-based metric and it was expected that Surface loss would achieve a high performance. However, as shown in Table 2, Dice loss has the lowest HD95, while Surface loss had the highest. One possible reason is that the Surface loss is relatively hard to train, requiring more epoches for it to converge. Since the training process was consistent across all loss functions, this may explain why some functions did not perform as well as expected.

To overcome variability in performance of individual loss functions, compound loss functions can be considered. For example, in the case of prostate segmentation, data imbalance is a major problem, and loss functions, such as BCE, that are suitable for balanced data are not suitable for this task. However, as shown in our study, weighted BCE combined with Dice can improve model performance significantly.

Tuning hyper-parameters of U-Net, such as the learning rate and number of iterations, requires significant computational time. To address this, we defined the best learning rate for Dice and BCE loss functions, as most of the other loss functions are variations of these loss functions. We used a grid search for optimization of the learning rate and defined the optimal value of loss function parameters in Focal, W (Dice + BCE) and Focal Tversky loss functions on the validation data set. The optimal learning rate was selected as α = 0.0001, from 0.001, 0.0001, 0.00001. The parameters of the W (Dice + BCE) loss function allocated a higher contribution to the cross-entropy term, α equal to 0.6, in comparison to the Dice term with a weight of 0.4. The optimum value of β for the weighted cross-entropy term was found to be 0.7, which penalizes false negatives more. This aligned with other recommendations for segmentation problems on MRI data [24]. Different values of α and β can be applied to obtain the best model result and handle the imbalance problem of each dataset appropriately.

Models were trained using the T2w axial data and performed better visually in the axial view. Training a model using axial, sagittal, and coronal (or a 3D data set) might improve the model performance. However, adding more inputs will also add complexity and extra computation cost. In this study, we used the 2D U-Net model, which has a lower number of components, to optimize in comparison to a 3D U-Net. In addition, 3D U-Net models underfit when trained on a small number of datasets [6]. Furthermore, it is easier to identify the loss function contribution to the model performance where there is less model complexity. It has been shown that a simple network with a proper loss function can outperform more complex architectures, including networks with specific up-sampling or with skip connection [24].

Regarding implementation, Keras offers a number of tools to construct a U-Net with its sequential and functional interface. Hence, the model itself can be constructed and set up for training in a straightforward approach. However, for the loss function, a potential challenge is to carefully choose the exact equation to implement. This is because even for the same loss function, there are slight variations. For example, the denominator of a Dice loss can be the sum of squared signal intensities, while another form will leave out the square operation. Such subtle differences can add to confounding factors when comparing model performance reported in the literature.

A model’s output can improve using post-processing methods that reduce false positives and false negatives in segmented images [25]. CNN segmentation results improve using energy-based refinement post-processing steps [26]. We applied threshold-based refinement to cope with false positives [27]. A threshold value of 0.5 was found to be the optimal value to return the highest Dice score with the least number of false positives.

## 5. Conclusions

The performance of a 2D U-Net model with nine different loss functions for prostate gland segmentation was compared. Ranking of model performance was found to depend on the metric used to measure performance. Performance was also found to vary based on the region within the prostate being considered, with the base and apex generally being less compared with the mid-glad and entire prostate gland. There was some evidence that performance was also affected by cross-sectional area of the image, with peak performance in the range of 600–2100 mm^2^. The performance of models using different loss functions varied by approximately 34% using the DSC score metric. Focal Tversky, Tversky, and W (Dice + BCE) loss functions achieve better performance considering majority of metrics. However, performance of models with Focal Tversky, Tversky, W (Dice + BCE), Dice, and IoU were close. Lower performance was observed using the distribution-based and boundary-based loss functions (Surface, BCE, and Focal loss functions). Based on this 37-patient data set, it is suggested that the Focal Tversky and W (Dice + BCE) loss functions are most suitable for the task of prostate segmentation as their parameters allow the user to modify the loss function for a specific dataset.

## Figures and Tables

**Figure 1 bioengineering-10-00412-f001:**
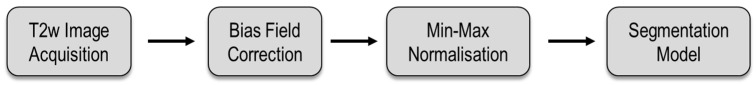
The image pre-processing for T2-weighted images. After acquisition, bias field correction was applied using the N4 algorithm to correct for the magnetic field inhomogeneity. The images were then normalized using the min–max approach before entering the segmentation network.

**Figure 2 bioengineering-10-00412-f002:**
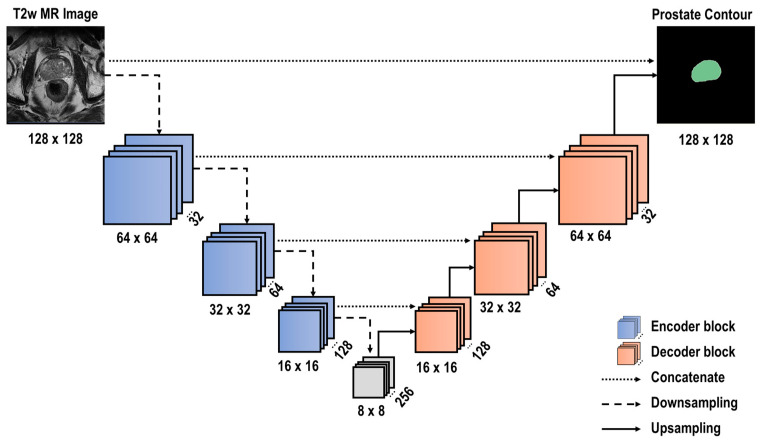
The U-Net architecture used in this study. The encoder contains four convolution layers with pooling. The decoder is symmetrical as the encoder, expanding the in-plane resolution back to the input image.

**Figure 3 bioengineering-10-00412-f003:**
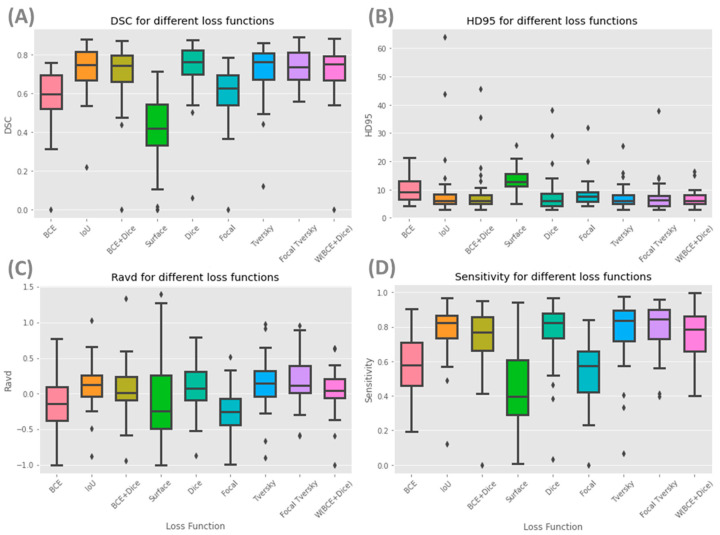
**Each** box plot (**A**–**D**) represents metrics DSC, HD95, Ravid and sensitivity respectively for the whole prostate on validation data from the five-fold cross-validation for models with different loss functions. DSC: Dice similarity coefficient; HD95: 95% Housdorff Distance.

**Figure 4 bioengineering-10-00412-f004:**
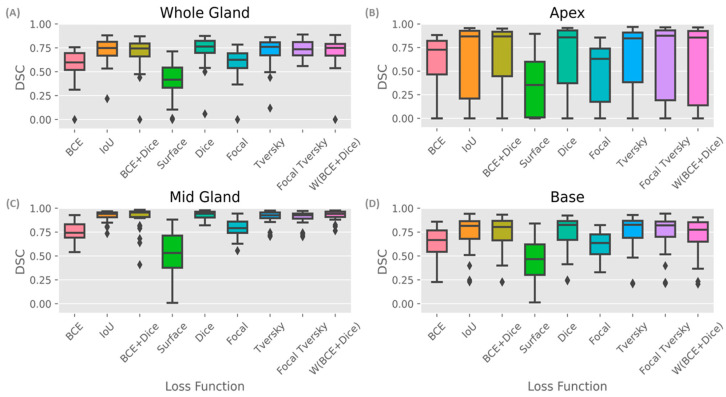
Boxplots showing the Dice similarity coefficient (DSC) scores for different parts of the prostate. The mid-gland (**C**) shows a consistent high performance (except for Surface loss), followed by the base (**D**) and the apex (**B**). The whole gland’s performance resembles the mid-gland (**A**,**C**), as it accounts for the majority of the prostate volume. Results are from the model trained using the Dice loss.

**Figure 5 bioengineering-10-00412-f005:**
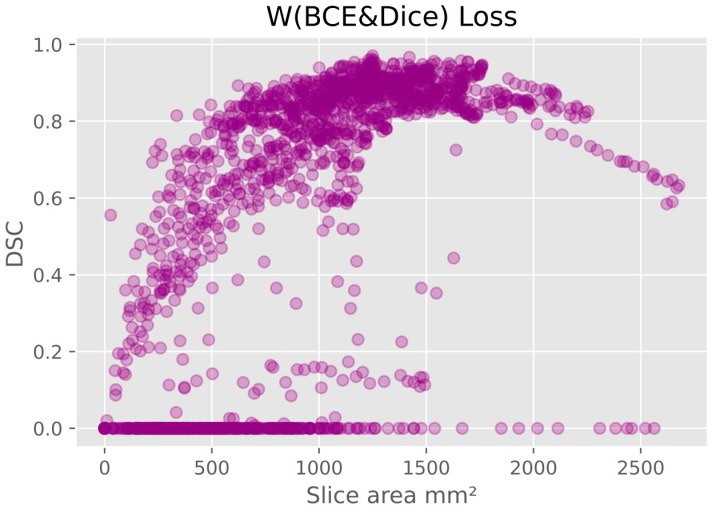
Dice score as a function of prostate area A reverse U shape is observed, indicating the prediction at the mid-gland (500–2000 mm^2^) outperformed those at the base (>2000 mm^2^) and apex (<500 mm^2^). The zeros at the bottom correspond to cases where the model totally missed the prostate region (Dice score = 0). Data is based on the prediction from the model using W (BCE + Dice) on the validation data. The same pattern is seen in all models. W (BCE + Dice): weighted binary cross-entropy with Dice.

**Figure 6 bioengineering-10-00412-f006:**
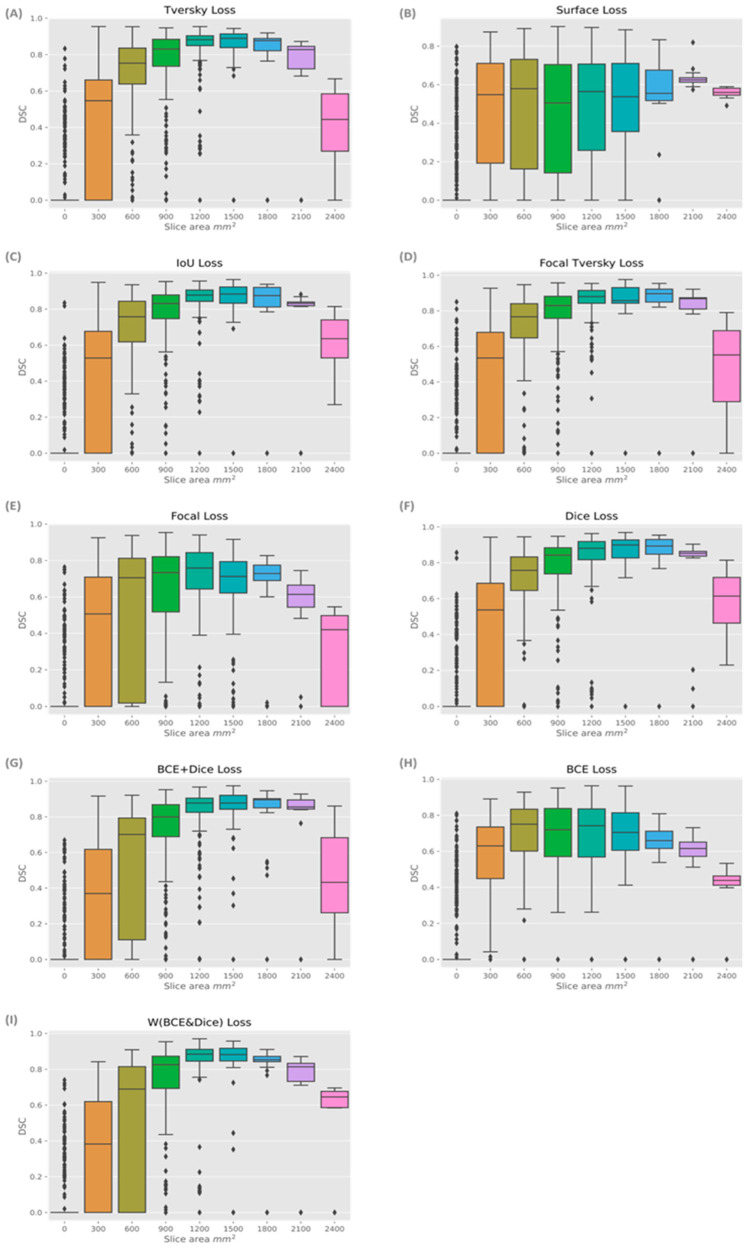
Box plots of DSC score vs. prostate area for each of the nine loss functions (**A**–**I**) listed in Table 1. DSC: Dice similarity coefficient.

**Figure 7 bioengineering-10-00412-f007:**
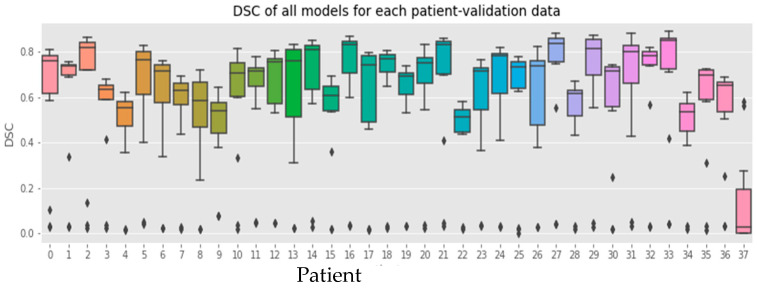
Box plots of Dice similarity coefficient (DSC) scores for all models for each patient in the validation data set.

**Figure 8 bioengineering-10-00412-f008:**
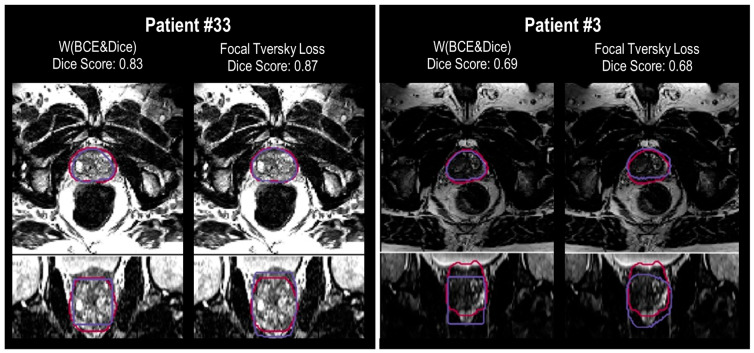
Outputs of two models (Focal Tversky and W (BCE + Dice)) in the axial (top image) and coronal views (bottom image for each patient), demonstrating both high and low performances, measured in Dice scores. The ground truth is represented by the red contour, the model’s prediction contour is shown in purple.

**Figure 9 bioengineering-10-00412-f009:**
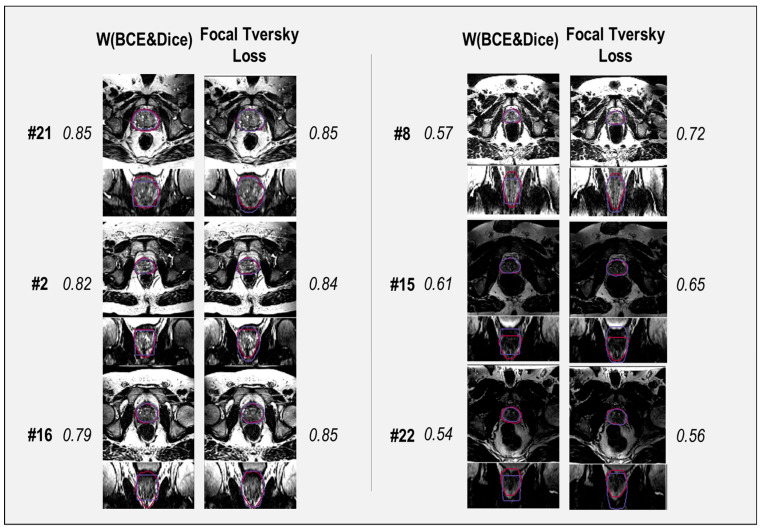
Outputs of two models (Focal Tversky and W (BCE + Dice)) for 6 patients (each patient identified by a number, e.g., #21 represents patient 21) in the axial (top image) and coronal views (bottom image for each patient), demonstrating both high and low performances, measured in Dice scores (shown adjacent to each image set). The ground truth is represented by the red contour, the model’s prediction contour is shown in purple.

**Table 1 bioengineering-10-00412-t001:** The family and individual loss functions used in this study.

Category	Loss Functions/Use Case
Distribution-based	**Binary CrossEntropy (BCE) Loss:** ○ Balanced dataset ○ Bernoulli distribution-based loss function **Focal Loss:** Suitable for highly imbalanced datasets ○ Enables models to learn hard examples by down-weighting simple samples
Region-based	**Intersection over Union (IoU) Loss** ○ Inspired from Jaccard similarity coefficient, a metric for segmentation validation **Dice Loss:** ○ Based on Dice coefficient **Tversky Loss:** ○ Variant of Dice coefficient ○ Adds weights to false positive and false negative **Focal Tversky Loss:** ○ Suitable for highly imbalanced dataset ○ Enables models to learn hard examples by down-weighting simple samples
Boundary-based	**Surface (Boundary) Loss**
Compound	**Weighted BCE and Dice W(BCE + Dice) Loss:** ○ Combination of Dice Loss and Binary CrossEntropy Loss ○ Used for lightly class imbalance ○ Benefits from both BCE and Dice Loss properties **BCE and Dice:** (BCE + Dice) Loss

**Table 2 bioengineering-10-00412-t002:** Mean value of the five-fold cross validation for each metric used in the current study are shown for the whole gland, base, mid-gland, and apex regions. Values shown in **bold** represent best performing results.

Loss Function	Part	BCE	BCE + Dice	Dice	Focal	Focal Tversky	IoU	Surface	Tversky	W (BCE + Dice)
DSC (mean± std)	Whole	0.58 ± 0.15	0.71 ± 0.16	**0.73** ± 0.15	0.60 ± 0.14	**0.74** ± 0.09	**0.73** ± 0.12	0.40 ± 0.18	**0.72** ± 0.14	0.71 ± 0.15
	Base	0.65 ± 0.16	0.73 ± 0.19	**0.74** ± 0.18	0.62 ± 0.15	**0.75** ± 0.19	**0.74** ± 0.18	0.46 ± 0.22	**0.74** ± 0.18	0.72 ± 0.19
	Mid	0.75 ± 0.11	0.90 ± 0.12	**0.93** ± 0.05	0.79 ± 0.09	0.90 ± 0.07	**0.92** ± 0.06	0.52 ± 0.24	0.90 ± 0.07	**0.93** ± 0.05
	Apex	0.59 ± 0.31	**0.65** ± 0.36	**0.65** ± 0.36	0.50 ± 0.31	0.63 ± 0.39	**0.64** ± 0.38	0.38 ± 0.32	0.63 ± 0.37	0.62 ± 0.39
95HD (mean± std)	Whole	10.41 ± 4.51	8.63 ± 8.29	**7.99** ± 7.05	8.54 ± 4.87	**7.42** ± 5.81	9.48 ± 11.37	13.64 ± 4.38	**7.17** ± 4.21	**6.66** ± 2.82
	Base	4.88 ± 1.62	4.51 ± 2.47	**4.41** ± 2.99	4.69 ± 1.51	5.11 ± 6.25	7.83 ± 12.71	9.47 ± 4.33	**4.22** ± 2.05	4.64 ± 2.39
	Mid	3.30 ± 1.13	1.65 ± 1.11	**1.49** ± 0.65	2.73 ± 0.76	1.85 ± 0.89	1.54 ± 0.73	5.52 ± 2.26	1.74 ± 0.85	1.51 ± 0.66
	Apex	4.12 ± 2.32	3.01 ± 2.43	**2.89** ± 2.45	4.04 ± 1.96	**2.97** ± 2.63	3.06 ± 2.56	5.63 ± 2.61	3.25 ± 2.79	3.13 ± 2.54
Ravd	Whole	−0.13 ±0.39	**0.07** ± 0.37	0.09 ± 0.31	−0.25 ± 0.31	0.18 ± 0.35	0.13 ± 0.32	−0.09 ± 0.62	0.15 ± 0.36	**0.05** ± 0.31
	Base	**−0.06** ± 0.93	0.69 ± 1.50	0.58 ± 1.47	**0.00** ± 0.98	0.73 ± 1.74	0.64 ± 1.51	0.21 ± 1.91	0.65 ± 1.66	0.70 ± 1.63
	Mid	−0.32 ± 0.27	**0.03** ± 0.25	**0.04** ± 0.15	−0.28 ± 0.21	0.15 ± 0.26	0.06 ± 0.20	−0.55 ± 0.26	0.09 ± 0.26	0.06 ± 0.17
	Apex	1.63 ± 4.64	1.23 ± 4.05	1.30 ± 3.72	**0.70** ± 2.98	1.89 ± 5.14	1.33 ± 3.97	**−0.31** ± 0.94	2.06 ± 5.76	1.7 ± 5.0
Sensitivity	Whole	0.58 ± 0.17	0.74 ± 0.21	0.77 ± 0.20	0.54 ± 0.19	**0.80** ± 0.17	0.78 ± 0.23	0.44 ± 0.28	0.78 ± 0.20	0.76 ± 0.05
	Base	0.60 ± 0.001	0.87 ± 0.01	0.85 ± 0.01	0.59 ± 0.001	**0.88** ± 0.01	0.86 ± 0.01	0.48 ± 0.01	0.86 ± 0.01	0.84 ± 0.01
	Mid	0.64 ± 0.17	0.92 ± 0.17	0.95 ± 0.06	0.68 ± 0.14	**0.96** ± 0.04	0.94 ± 0.06	0.41 ± 0.24	0.94 ± 0.06	0.95 ± 0.05
	Apex	0.74 ± 0.17	0.87 ± 0.21	0.87 ± 0.20	0.66 ± 0.19	**0.90** ± 0.17	0.84 ± 0.23	0.45 ± 0.28	0.87 ± 0.20	**0.95 ± 0.05**
Precision	Whole	0.69 ± 0.14	0.71 ± 0.19	0.72 ± 0.14	**0.73** ± 0.17	0.71 ± 0.14	0.72 ± 0.13	0.51 ± 0.16	0.71 ± 0.13	**0.73** ± 0.13
	Base	**0.83** ± 0.22	0.69 ± 0.24	0.73 ± 0.24	0.79 ± 0.23	0.72 ± 0.25	0.72 ± 0.25	0.64 ± 0.33	0.72 ± 0.24	0.70 ± 0.26
	Mid	**0.96** ± 0.09	0.91 ± 0.09	0.92 ± 0.08	**0.96** ± 0.07	0.86 ± 0.13	0.91 ± 0.10	0.92 ± 0.11	0.89 ± 0.13	0.91 ± 0.09
	Apex	0.77 ± 0.34	0.77 ± 0.27	**0.79** ± 0.30	0.78 ± 0.31	0.75 ± 0.32	0.78 ± 0.30	0.77 ± 0.26	0.75 ± 0.32	0.71 ± 0.30

## Data Availability

The data presented in this study and the code used to perform the calculations with these data are available on request from the corresponding author. The data are not publicly available due to privacy restrictions.

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
