# Peer review of "U-Net Architecture for Prostate Segmentation: The Impact of Loss Function on System Performance"

_bioengineering, 2023, doi:10.3390/bioengineering10040412_

Round 1

Reviewer 1 Report

This paper presents a comparative study of loss functions for prostate segmentation on MR images using the U-Net architecture. The idea is very interesting but needs some revisions.

1. Try to improve the title. In my opinion, it does not look interesting (Suggestion)

2. Introduction part needs to be improved.

3. Abstract should be to the point.

4. Add some relevant literature relevant to your methodology. 

5. The paper used 5-fold cross-validation, but there are no references or discussion in the literature: 
i. Khan, J., Fayaz, M., Hussain, A., Khalid, S., Mashwani, W. K., & Gwak, J. (2021). An improved alpha beta filter using a deep extreme learning machine. IEEE Access9, 61548-61564.

ii. Khan, J., Lee, E., & Kim, K. (2022). A higher prediction accuracy–based alpha–beta filter algorithm using the feedforward artificial neural network. CAAI Transactions on Intelligence Technology.

iii. Khan, J., & Kim, K. (2022). A Performance Evaluation of the Alpha-Beta (α-β) Filter Algorithm with Different Learning Models: DBN, DELM, and SVM. Applied Sciences12(19), 9429.

the following papers used a cross-validation model.

6. please add future research direction in your conclusion part.

Reviewer 2 Report

A comparative study of nine loss functions for prostate segmentation on MR images using the U-Net architecture is studied in this paper. It is interesting and quite meaningful to systematically explore those loss functions on a specific application, here is prostate segmentation. My comments are:

1)      There are many loss functions, it is necessary to explain more why those nice loss functions are chosen?

2)      The dataset used in the study are not public dataset, is there public dataset available for prostate segmentation? If yes, why not also including those datasets? As the results maybe different for different domain’s data

3)      There are some work trying to explore the effect of loss functions, such as ‘Loss odyssey in medical image segmentation’, which studied 20 loss functions on six public datasets from medical centers. It would be necessary to better present the contributions of this work to the community.

4)      Fig.3 shows the box plot for W loss, how about others? Besides, the visualization of the results can be improved.

5)      The difference between model performance was approximately 34% in Dice similarity coefficient (DSC) score, how about the others? Are the performance be influenced this much also? Why?

Reviewer 3 Report

In this paper, authors examined the effect of loss functions on the performance of deep learning-based prostate segmentation models. A U-Net model for prostate segmentation using T2-weighted images from a local dataset was trained and performance compared when using nine different loss functions. The following review comments are recommended, and the authors are invited to explain and modify.

Comment: In abstract authors claimed “The performance and accuracy of deep learning models varies depending to the design and optimal tuning of the hyper-parameters”, later just told, used grid search for optimisation of the learning rate, how they used it, need explanation.

Comment: “Pre-processing of the input data”, I think, authors should show some Figures for better understanding.   

Comment: “Prostate segmentation using U-Net have several fundamental limitations”, but still U-net is a better one for medical image segmentation. What are these fundamental limitations in U-Net, and how did authors research on that with some possible solutions?

Comment: Authors did not mention implementation challenges.

Comment: When writing phrases like “Models were compared and evaluated using the Dice coefficient”, it must cite some related work in order to sustain the statement (10.3390/math10050796).

Comment: “We applied threshold based refinement to cope with false positives”, need explanation.

Comment: Could you please check your references carefully? All references must be complete before the acceptance of a manuscript.

Reviewer 4 Report

The manuscript submitted by Montazerolghaem and co-authors analyses the impact of different loss functions for the specific case of segmentation of prostate from MR images. The concept of the paper is fairly straight forward and very relevant, as many people are using the U-Net architecture and, as shown in the manuscript, the use of a certain loss function can have a huge impact in the results. As such, I am confident to recommend for publication on the basis of the interest and results. However, the manuscript can be improved in terms of readability and clarity.

Comments:

11)     The justification (lines 98 and 99) on the selection of these loss functions is not particularly strong, which are traditional (with references) and which are more recent (again with references). Then 100,101 states “Most of the loss functions used in this study were selected based on their suitability for 100 use with strongly and mildly imbalanced data sets in segmentation tasks.” If these were the selected ones, which ones were not selected? Until when were loss functions considered? There are quite a few loss functions being proposed, e.g.:

https://doi.org/10.1186/s12938-021-00937-w

https://doi.org/10.1016/j.media.2022.102509

https://doi.org/10.1109/jbhi.2022.3222390

https://doi.org/10.3390/s21082803

https://doi.org/10.1021/acs.analchem.1c02830

https://doi.org/10.1016/j.eswa.2022.118833

It is fair to compare just a few loss functions, but a strong justification should be presented and a comment on why some others were not included could also go in the discussion as further work.

22)     All figures should be labelled so that they can be referred, e.g. Fig 2a, Fig 2b, etc. This will be particularly relevant in points below.

33)     Table 2 is very important and a bit difficult to read. Why are some values in red? Since the columns have a lot of free space, why not make them smaller to fit the width of the text? It would help significantly is the best results were highlighted in Bold or underlined.

44)     Captions are rather poor. What we see is just a title of what is being shown, but the caption should make the figure self explanatory, that is, an explanation should give some insight into the data of the figure. What should a reader make of the scatter plots of Fig 2? What should we focus on? In particular for Fig 2 I would at least want to know why are there so many zeros and why there seems to be like a tail towards the right I some cases but not in others. I could make an educated guess (I think I know the answer) but it is the job of the authors not to let the readers to make educated guesses.

55)     Please be consistent in presentation. Horizontal axis of Fig 2 is slice area mm, Fig 3 is area (in mm? of slices?) and S6 is volume and they all have different ticks. This is a few lines of code in python/matlab/r to have everything consistent.

66)     Fig 4 is incredibly difficult to follow and again, I have to make educated guesses due to bad captions and figure structure. What is #21?What are the 3 images of one of these cases? Again, I can guess that we are talking about axial, coronal and sagittal planes, but maybe not. And if you want to illustrate differences in results, why start with one which gives the same values?

77)     I personally prefer papers that do not have supplementary materials. Supplementary materials make sense for Journals that have a restricted number of pages, but as far as I know, this is not the case for bioengineering. Thus, I would ask, if the results are relevant, add to the main body. If they are not, then we can skip them. I personally think that these are important and could be summarised, for instance, take the Dice boxplots for whole, mid and apex and add as a figure. That would be certainly better than Fig 3 which is rather difficult to interpret.

88)     I find the supplementary Dice results (let’s call them FS1a, FS2a, FS3a) the most interesting results. What we want is to know which loss function to select and these are good indication: BCE, Surface and Focal score below all other techniques, simple message. Then please explain (in the caption) the outliers and why is surface so low.

Round 2

Reviewer 1 Report

I have no more Comments.

Author Response

Please find attached comments in response to Reviewers 2 & 4 who requested further changes to our manuscript.

Reviewer 2 Report

Thanks for the revision. It seems the comments previously mentioned is not well addressed. 

The previous comments and the new questions:

2) 'We agree that the performance may vary with different datasets, however, when comparing results across each of the loss functions, we do not believe the conclusions will change when considering their performance in the task of prostate autosegmentation.' It is necessary to proven this kind of statement

3) 'It would be necessary to better present the contributions of this work to the community.' Cite the paper is not that much relavant to this question

5) The loss functions can influence those much of the performance, it would be better to analyze it. "Are the other performance be influenced this much also? Why?"

Reviewer 3 Report

The authors have answered my questions satisfactorily.

Author Response

(The authors gave the same response as above.)

Reviewer 4 Report

The authors have improved the manuscript, however, from the comments I made, they only took into account about half of them. The introduction still does not present a strong justification of why these loss functions were selected, the figures still have area / slice area inconsistencies, the figures do not use 2a, 2b, 2c, etc. and a few more comments were not properly addressed.

The recommendation of the loss function in the discussion is based on the results obtained, but this is different from a strong justification of why these particular loss functions were selected. That is part of the introduction.

As such I cannot recommend for publication yet.

Round 3

Reviewer 2 Report

2.2, I am not asking to cite the paper, which is already cited in the previous version. What I mean is, those loss functions are studies already, not only that paper. This paper again used those loss functions with another application. It would be better to clarify what's the new fundings.

Reviewer 4 Report

The authors have addressed all my concerns